# Uncertainty-Aware Generative Oversampling Using an Entropy-Guided Conditional Variational Autoencoder

## Abstract

Class imbalance remains a major challenge in machine learning, especially for high-dimensional biomedical data where nonlinear manifold structures dominate. Traditional oversampling methods such as SMOTE rely on local linear interpolation, often producing implausible synthetic samples. Deep generative models like Conditional Variational Autoencoders (CVAEs) better capture nonlinear distributions, but standard variants treat all minority samples equally, neglecting the importance of uncertain, boundary-region examples emphasized by heuristic methods like Borderline-SMOTE and ADASYN. We propose Local Entropy-Guided Oversampling with a CVAE (LEO-CVAE), a generative oversampling framework that explicitly incorporates local uncertainty into both representation learning and data generation. To quantify uncertainty, we compute Shannon entropy over the class distribution in a sample's neighborhood: high entropy indicates greater class overlap, serving as a proxy for uncertainty. LEO-CVAE leverages this signal through two mechanisms: (i) a Local Entropy-Weighted Loss (LEWL) that emphasizes robust learning in uncertain regions, and (ii) an entropy-guided sampling strategy that concentrates generation in these informative, class-overlapping areas. Applied to clinical genomics datasets (ADNI and TCGA lung cancer), LEO-CVAE consistently improves classifier performance, outperforming both traditional oversampling and generative baselines. These results highlight the value of uncertainty-aware generative oversampling for imbalanced learning in domains governed by complex nonlinear structures, such as omics data.

## 1 Introduction

The class imbalance problem, characterized by a severely skewed distribution of samples across classes, remains a critical challenge in machine learning (Chen et al., 2024). Standard learning algorithms, optimized for overall accuracy, tend to develop a strong predictive bias towards the majority class (Das et al., 2018). Consequently, instances from the minority class, which are often of greatest interest in high-stakes domains like medical diagnosis, fraud detection, and industrial fault prediction, are frequently misclassified (Buda et al., 2018; Makki et al., 2019; Malhotra & Kamal, 2019).

To mitigate this issue, a variety of techniques have been developed, broadly categorized into algorithm-level and data-level approaches (Gao et al., 2025; Yang et al., 2024; Buda et al., 2018). Among these, data-level oversampling methods are particularly popular due to their model-agnostic nature (Chen et al., 2024; Buda et al., 2018). The simplest approach, Random Oversampling (Japkowicz, 2000), duplicates minority samples but risks severe overfitting. A more sophisticated alternative, the Synthetic Minority Oversampling Technique (SMOTE) (Chawla et al., 2002), generates new samples via linear interpolation between neighboring minority instances. SMOTE has become a foundational baseline and has inspired a wide range of variants (Fernández et al., 2018; Douzas et al., 2018; Douzas & Bação, 2019; Kunakorntum et al., 2020; Li et al., 2025a; Wang et al., 2025b). For example, Borderline-SMOTE (Han et al., 2005) and ADASYN (Haibo et al., 2008) direct synthetic generation toward minority samples located near class boundaries, reflecting the intuition that these "hard-to-learn" points are more critical for classification.

Despite these advances, traditional oversampling methods share critical weaknesses. Their reliance on local, linear interpolation restricts the diversity of generated data, often confining synthetic points within the convex hull of minority samples (Dai et al., 2019; Wang et al., 2025a). Moreover, because they rely solely on minority neighbors, they ignore the global data structure and the informative role of the majority distribution, which frequently results in noisy samples in regions of class overlap (Batista et al., 2004; Ai et al., 2023).

These shortcomings have motivated a paradigm shift towards deep generative models, which can learn global data distributions to generate novel and consistent samples (Liu et al., 2007). While Denoising Diffusion Models have achieved state-of-the-art performance in high-fidelity image synthesis (Ho et al., 2020), their application to tabular data remains a challenging area of research due to complex, non-Gaussian feature distributions (Li et al., 2025b). For tabular settings, they are also computationally intensive to train and sample from (Shi et al., 2025).

By contrast, Generative Adversarial Networks (GANs) (Goodfellow et al., 2014) and Variational Autoencoders (VAEs) (Kingma & Welling, 2013) have been more widely explored for tabular data generation. GANs, however, often suffer from training instability (Sampath et al., 2021), whereas VAEs provide a more stable and tractable alternative. Among them, the Conditional VAE (CVAE) (Sohn et al., 2015) offers a principled framework for class-conditioned oversampling and has been successfully applied in imbalanced learning (Fajardo et al., 2021). However, a key limitation of standard CVAEs is that they are agnostic to sample importance, treating all minority instances as equally informative. This uniform approach overlooks a critical insight from heuristic methods like Borderline-SMOTE (Han et al., 2005) and the Adaptive Synthetic Sampling (ADASYN) (Haibo et al., 2008): not all samples are equally valuable for refining a classifier's decision boundary. Instances located deep within a class's feature space are less informative than those situated in regions of high class overlap, where the boundary is most ambiguous. These "borderline" or "hard-to-learn" samples are strategically crucial, as they provide the most challenging examples for a classifier to learn (Japkowicz & Stephen, 2002; Haibo et al., 2008).

Recent research has extended CVAEs for imbalanced learning along several key axes: (i) *loss-function modification*, which applies focal-style losses to the reconstruction objective to emphasize hard-to-reconstruct samples (Lin et al., 2017); (ii) *latent space structuring*, as in DVAE (Guo et al., 2019) and CTVAE (Wang et al., 2025a), which engineer a more discriminative latent space to generate boundary-focused samples; and (iii) *knowledge transfer*, as in MGVAE (Ai et al., 2023), which uses transfer learning from the majority class to learn robust representations for the minority class. While these approaches improve generation quality, they either reweight reconstruction, engineer latent geometry, or borrow majority information, and none directly address sample-level uncertainty.

Consequently, a significant gap remains in integrating this principle of uncertainty-awareness into the powerful distributional learning capabilities of a deep generative model. This gap is particularly pronounced in clinical genomics, which is the central focus of the current study. We hypothesize that the high dimensionality and intricate, nonlinear relationships in this domain cause the core assumption of linear interpolation used by SMOTE to break down, resulting in biologically implausible synthetic data (Blagus & Lusa, 2012). Furthermore, the inherent biological heterogeneity and the continuum-like nature of disease progression mean that class boundaries are rarely sharply defined, creating regions of high predictive uncertainty for a classifier. To leverage this insight, we turn to information theory to develop a formal measure of this local uncertainty. We quantify the degree of class overlap using Shannon entropy, the canonical measure of uncertainty, allowing us to identify high-entropy regions as a quantitative proxy for a sample's 'hard-to-learn' status.

In essence, this combination of a complex, non-linear data manifold and inherently ambiguous class boundaries establishes clinical genomics as a uniquely challenging domain. This setting reveals the limitations of two key approaches: traditional oversampling methods, which struggle with complex, non-linear data, and standard generative models, which are agnostic to uncertain and hard-to-learn regions in the feature space. Consequently, it provides an ideal setting to validate a generative framework specifically engineered to target and learn from these high-uncertainty regions.

To this end, we propose the Local Entropy-Guided Oversampling with a CVAE (LEO-CVAE). We formalize the notion of a "hard-to-learn" or "uncertain" region using local Shannon entropy, a measure that quantifies the mixture of class labels within a sample's local neighborhood. The LEO-CVAE framework leverages this local entropy signal in two synergistic ways: first, it guides the CVAE's

training process through a weighted loss function that prioritizes learning in the high-entropy regions; second, it directs the synthetic data generation by preferentially selecting high-entropy instances as seeds. By concentrating both the learning and generative processes on these ambiguous areas, LEO-CVAE directly reinforces the classifier's decision boundary where it is weakest. The primary contributions of this work are threefold: (i) a **Novel Uncertainty Metric**, where we introduce the Local Entropy Score (LES) to formally quantify sample-level uncertainty, identifying the most informative, class-overlapping regions for oversampling; (ii) an **Uncertainty-Aware Generative Framework**, where we propose LEO-CVAE, which integrates the LES signal through two core components: a Local Entropy-Weighted Loss (LEWL) and an entropy-guided sampling strategy; and (iii) **Empirical Validation**, where we demonstrate the effectiveness of LEO-CVAE on challenging imbalanced clinical genomics datasets for both binary and multiclass classification through a systematic comparison against a suite of traditional and generative oversampling methods.

## 2 METHODS

This section details our proposed oversampling method, Local Entropy-Guided Oversampling with a Conditional Variational Autoencoder (LEO-CVAE). We first provide the problem formulation, followed by an overview of the CVAE, which serves as our generative foundation. We then introduce our core contribution: the Local Entropy Score (LES), a metric for quantifying sample-level uncertainty to identify high-entropy regions within the feature space. Finally, we describe how LES is integrated into the CVAE through our two novel mechanisms: the Local Entropy-Weighted Loss (LEWL) for model training and the Entropy-Guided Sampling strategy for data generation.

### 2.1 PROBLEM FORMULATION

Let $\mathcal{D} = \{(x_i, y_i)\}_{i=1}^{N}$ be a training dataset of $N$ samples, where $x_i \in \mathbb{R}^D$ is a $D$-dimensional feature vector and $y_i \in \{c_1, c_2, \ldots, c_C\}$ is its corresponding class label. The dataset $\mathcal{D}$ is imbalanced if the class distribution is skewed, i.e., there exists a majority class $c_{maj}$ and a minority class $c_{min}$ such that the number of samples $N_{maj} \gg N_{min}$. The goal of oversampling is to generate a new set of synthetic minority samples, $\mathcal{D}_{syn}$, such that when combined with the original data ($\mathcal{D}' = \mathcal{D} \cup \mathcal{D}_{syn}$), the resulting dataset is balanced or near-balanced, leading to improved performance of a classifier trained on $\mathcal{D}'$.

### 2.2 CVAE FOUNDATION

Our method is built upon a CVAE (Sohn et al., 2015), a generative model that learns a latent representation of data conditioned on class labels. A CVAE consists of two neural networks: an encoder and a decoder.

The encoder network, parameterized by $\phi$, learns to approximate the intractable true posterior distribution $p(z|x, c)$. It maps a data point x and its class condition c to the parameters of a diagonal Gaussian distribution, $q_\phi(z|x, c) = \mathcal{N}(z|\mu, \text{diag}(\sigma^2))$. The mean vector $\mu$ and variance vector $\sigma^2$ are the direct outputs of the encoder network.

The decoder network, parameterized by $\theta$, reconstructs the data by modeling the distribution $p_\theta(x|z, c)$. To generate a reconstructed sample $\hat{x}$, a latent vector z is first sampled from the encoder's output distribution using the reparameterization trick: $z = \mu + \epsilon \odot \sigma$, where $\epsilon \sim \mathcal{N}(0, I)$. This sample z is then concatenated with the one-hot class vector $c_{oh}$ and passed as input to the decoder.

The standard CVAE is trained by minimizing a loss function derived from the negative of the Evidence Lower Bound (ELBO):

$$\mathcal{L}_{\text{CVAE}} = -\mathbb{E}_{q_\phi(z|x,c)}[\log p_\theta(x|z, c)] + \beta \cdot D_{KL}(q_\phi(z|x, c)||p(z|c)) \tag{1}$$

The first term is the reconstruction loss, which measures how well the model reconstructs the input data, and the second is the Kullback-Leibler (KL) divergence, which regularizes the latent space to follow a prior distribution $p(z|c)$.

## 2.3 LOCAL ENTROPY SCORE (LES)

The core novelty of our method is the introduction of LES to guide the CVAE. LES quantifies the complexity of the feature space surrounding a given sample. For each sample $x_i$, we first identify its $k$ nearest neighbors, $\mathcal{N}_k(x_i)$. The neighborhood is determined using the standard Euclidean distance between feature vectors. We then compute the probability distribution of classes within this neighborhood. Let $k_j$ be the count of neighbors belonging to class $c_j$. The local probability of class $c_j$ is $p(c_j|x_i) = k_j/k$. The LES for sample $x_i$, denoted as $H(x_i)$, is then calculated using the Shannon entropy formula:

$$H(x_i) = -\sum_{j=1}^{C} p(c_j|x_i) \log_2 p(c_j|x_i) \tag{2}$$

For this calculation, we follow the standard convention that the contribution of any class $c_j$ with $p(c_j|x_i) = 0$ is taken to be 0, as $\lim_{p \to 0^+} p \log p = 0$.

The resulting score, which we denote $H_i = H(x_i)$, provides a quantitative measure of the heterogeneity of the sample's local feature space. The score ranges from a minimum of 0, signifying a perfectly homogenous neighborhood where all neighbors belong to the same class, to a theoretical maximum of $\log_2(C)$, where $C$ is the total number of classes. This maximum value corresponds to a state of maximum entropy, representing the highest possible uncertainty where neighbors are uniformly distributed across all classes. A high LES, therefore, directly identifies a sample located in a complex, class-overlapping region of the feature space. This score becomes the critical signal for guiding both the learning and generation processes in our LEO-CVAE framework.

## 2.4 THE LEO-CVAE METHOD

LEO-CVAE leverages the calculated local entropy scores in two critical stages: modifying the CVAE loss function to focus learning on high-entropy regions and guiding the generation of new synthetic samples.

### 2.4.1 THE LOCAL ENTROPY-WEIGHTED LOSS (LEWL)

To guide the CVAE's training, our primary innovation is a novel loss function, the LEWL. It instills uncertainty-awareness by modifying the CVAE's reconstruction component, while the standard KL divergence term is retained for its conventional regularization role. The complete LEWL objective is a composite function defined as:

$$\mathcal{L}_{\text{LEWL}} = \mathcal{L}_{\text{W-Recon}} + \beta \cdot \mathcal{L}_{\text{KLD}} \tag{3}$$

Here, $\beta$ is a hyperparameter that weights the contribution of the Kullback-Leibler (KL) divergence term. We detail each component below.

**Weighted Reconstruction Loss ($\mathcal{L}_{\text{W-Recon}}$):** The innovation of our training paradigm is captured in this component. We replace the CVAE's standard reconstruction error with a Weighted Mean Squared Error (MSE). This loss component strategically prioritizes samples that are most informative for learning a robust model: those belonging to minority classes and those located in complex, class-overlapping regions identified by a high LES. The loss for a single sample $x_i$ with class label $y_i$ and pre-calculated local entropy $H_i$ is defined as:

$$\mathcal{L}_{\text{W-Recon},i} = w_{\text{class}}(y_i) \cdot w_{\text{entropy}}(H_i) \cdot \|x_i - \hat{x}_i\|_2^2 \tag{4}$$

The two weighting factors are:

- **Class-Imbalance Weight ($w_{\text{class}}$):** This weight is calculated from the inverse frequency of each class in the training data. By assigning a higher weight to samples from underrepresented classes, it compels the model to overcome the inherent bias of the imbalanced dataset.

- **Entropy-Focus Weight ($w_{\text{entropy}}$):** This factor directs the model's attention toward samples in ambiguous, high-entropy regions. It is defined as $w_{\text{entropy}}(H_i) = (1 + H_i)^{\gamma}$, where the hyperparameter $\gamma \geq 0$ controls the intensity of the focus. When $\gamma = 0$, this guidance is inactive. For $\gamma > 0$, the model is more heavily penalized for failing to accurately

reconstruct samples located in high-entropy regions, forcing it to learn a more discriminative representation of the decision boundary.

The final reconstruction loss, $\mathcal{L}_{\text{W-Recon}}$, is the mean of these individually weighted losses calculated across all samples in a mini-batch.

**KL Divergence Loss ($\mathcal{L}_{\textbf{KLD}}$):** To ensure the latent space remains well-structured, we include the standard KL Divergence Loss. This is the unmodified, conventional regularization term from the CVAE's ELBO. It measures the divergence between the encoder's output distribution and a class-conditional prior and is defined as:

$$\mathcal{L}_{\text{KLD}} = D_{\text{KL}}(q_\phi(z|x,c)||p(z|c)) \tag{5}$$

To counteract the common CVAE training issue of posterior collapse, where the KLD term vanishes and the decoder learns to ignore the latent code $z$, we enforce a minimum threshold on this loss component during optimization. This ensures that the latent variables continue to encode meaningful information throughout the training process.

### 2.4.2 ENTROPY-GUIDED SAMPLE GENERATION

After training the LEO-CVAE, we use it to generate synthetic samples for each minority class until it reaches parity with the majority class. Applying the same core principle used in the LEWL, we focus the creation of new samples on high-entropy regions. The procedure for a given minority class $c_{min}$ is as follows:

1. **Calculate Generation Count:** $N_{gen} = N_{maj} - N_{min}$ (number of new samples).

2. **Select Seed Samples:** The seeds for generation are chosen from the original minority class instances in the training set. To prioritize the synthesis of new data in high-entropy regions, we employ a non-uniform selection strategy guided by LES. A sampling probability distribution, $P$, is established over the minority class samples where, instead of uniform selection, the probability of selecting a given sample $x_i$ is made proportional to its entropy-focus weight, the same transformation of its LES used in the LEWL:

$$P(x_i) \propto (1 + H_i)^\gamma, \quad \text{for all } (x_i, y_i) \text{ where } y_i = c_{\min} \tag{6}$$

The hyperparameter $\gamma$ again controls how strongly this selection process favors samples in high-entropy regions. From this entropy-weighted distribution, $N_{\text{gen}}$ seed samples are drawn with replacement to initiate the data generation process. This ensures that samples residing in high-entropy neighborhoods are more likely to be chosen as templates for creating new synthetic data.

3. **Generate Synthetic Data:** For each selected seed sample $x_{\text{seed}}$, a new synthetic sample $\hat{x}_{\text{new}}$ is generated. This is achieved by first encoding the seed to its latent distribution, then sampling a new latent vector $z_{\text{new}} \sim q_\phi(z|x_{\text{seed}}, c_{\min})$ using the reparameterization trick. Finally, the resulting vector $z_{\text{new}}$ is passed to the decoder to produce the synthetic sample $\hat{x}_{\text{new}}$.

This ensures that the synthetic data is generated around the most informative minority samples located in high-entropy regions, effectively reinforcing the decision boundary in contested regions of the feature space. The complete LEO-CVAE oversampling process is summarized in Algorithm 1.

## 3 EXPERIMENTAL SETUP

Experimental validation was conducted on two challenging clinical genomics datasets: The Cancer Genome Atlas (TCGA) lung cancer and Alzheimer's Disease Neuroimaging Initiative (ADNI). These datasets were specifically selected, as they are characterized by the high dimensionality and complex, non-linear relationships inherent to genomic data, necessitating a powerful generative model. Moreover, the inherent biological heterogeneity among patients and the gradual progression of disease mean that class boundaries are not sharp, distinct lines. Instead, these factors create a continuum where samples from different classes intermingle, forming complex, overlapping decision boundaries. This results in the exact high-entropy regions that our entropy-guided ('LEO') mechanism is designed to target and resolve, providing a perfect testbed to rigorously evaluate our method's capacity to handle these real-world clinical data challenges.

---

**Algorithm 1** LEO-CVAE Framework

---

**Input:** Training data $\mathcal{D} = \{(x_i, y_i)\}_{i=1}^N$, $k$ for k-nearest neighbors, hyperparameters $\gamma, \beta$.
**Output:** Resampled dataset $\mathcal{D}' = \mathcal{D} \cup \mathcal{D}_{syn}$.
    **Step 1: Quantifying Sample-Level Uncertainty**
1: **for** each sample $x_i \in \mathcal{D}$ **do**
2:     Find k-nearest neighbors $\mathcal{N}_k(x_i)$.
3:     Calculate local entropy score (LES) $H(x_i)$ using Eq. (2).
4: **end for**
    **Step 2: Training with Entropy-Weighted Loss**
5: Initialize LEO-CVAE model (Encoder$_\phi$, Decoder$_\theta$).
6: **for** number of training epochs **do**
7:     **for** each mini-batch $\{(x_b, y_b, H_b)\}_{b=1}^B$ **do**
8:         Compute $\mu_b, \log \sigma_b^2 = \text{Encoder}_\phi([x_b, c_{b\_oh}])$.
9:         Sample $z_b \sim \mathcal{N}(\mu_b, \text{diag}(\sigma_b^2))$.
10:        Reconstruct $\hat{x}_b = \text{Decoder}_\theta([z_b, c_{b\_oh}])$.
11:        Calculate loss $\mathcal{L}_{\text{LEWL}}$ using Eq. (3).
12:        Update $\phi$ and $\theta$ via gradient descent.
13:     **end for**
14: **end for**
    **Step 3: Entropy-Guided Generation**
15: $\mathcal{D}_{syn} \leftarrow \emptyset$
16: Identify minority classes $\mathcal{C}_{min}$ and majority count $N_{maj}$.
17: **for** each class $c_j \in \mathcal{C}_{min}$ **do**
18:     Let $\mathcal{D}_j$ be the set of samples in class $c_j$.
19:     Calculate sampling probabilities $P(x_i)$ for all $x_i \in \mathcal{D}_j$ using Eq. (6).
20:     $N_{gen} \leftarrow N_{maj} - |\mathcal{D}_j|$.
21:     Select $N_{gen}$ seed samples $\{x_{seed}\}$ from $\mathcal{D}_j$ with replacement using probabilities $P$.
22:     Generate $N_{gen}$ synthetic samples $\{\hat{x}_{new}\}$ from $\{x_{seed}\}$.
23:     $\mathcal{D}_{syn} \leftarrow \mathcal{D}_{syn} \cup \{(\hat{x}_{new}, c_j)\}$.
24: **end for**
25: **return** Resampled dataset $\mathcal{D}' = \mathcal{D} \cup \mathcal{D}_{syn}$.

---

## 3.1 DATASETS

**TCGA Lung Cancer Dataset:** Gene expression (RNA-Seq) and clinical data for lung adeno-carcinoma (LUAD) and lung squamous cell carcinoma (LUSC) were obtained from TCGA via UCSC Xena (Tomczak et al., 2015). A total of 817 patient samples with complete expression and Progression-Free Survival (PFS) records were included. From 17,738 gene expression features, mutual information–based selection yielded 64 informative ones. The classification task was to predict PFS, categorized as 'short' ($\leq 1$ year; 147 samples, minority class) or 'long' ($>1$ year; 670 samples, majority class).

**ADNI Dataset:** Blood-based gene expression data were obtained from the ADNI database (adni.loni.usc.edu) for a multiclass classification task. ADNI, launched in 2003 as a public–private partnership led by Principal Investigator Michael W. Weiner, was designed to assess whether imaging, biomarkers, and clinical/neuropsychological measures could track progression from mild cognitive impairment (MCI) to Alzheimer's disease (AD) (Petersen et al., 2010). Using mutual information–based feature selection, the initial 49,386 probe sets per sample were reduced to 64 features. Data from 744 participants (246 cognitively normal (CN), 382 MCI, and 116 AD) were analyzed for classification.

## 3.2 COMPARISON METHODS

We benchmarked our proposed LEO-CVAE against a diverse suite of seven comparison methods. These included standard baselines (No Oversampling; Random Oversampling (Japkowicz, 2000)), established heuristic methods (SMOTE (Chawla et al., 2002), Borderline-SMOTE (Han et al., 2005), ADASYN (Haibo et al., 2008)), and two generative models (CVAE (Sohn et al., 2015) and a CVAE

adapted with Focal Loss (Lin et al., 2017)). Full descriptions and specific configurations for each method are detailed in Appendix A.1 to ensure reproducibility.

### 3.3 EVALUATION PROTOCOL AND IMPLEMENTATION DETAILS

All experiments followed a consistent and rigorous evaluation protocol to ensure fairness and reproducibility.

**Experimental Design:** A 5-fold stratified cross-validation was employed. Oversampling methods were applied only to the training data, while validation sets remained untouched for unbiased performance estimates. A fixed random seed of 42 was used throughout.

**Oversampling Models and Baselines:** To ensure a fair comparison, baseline models were configured with consistent parameters. All k-NN–based methods (SMOTE, Borderline-SMOTE, ADASYN) used identical neighbor settings. Similarly, our proposed LEO-CVAE and all CVAE-based baselines shared the same network architecture and training parameters. Full details on architectures and hyperparameters are provided in Appendix A.1 and Appendix A.2.

**Downstream Classifier:** A Multi-Layer Perceptron (MLP) with a fixed architecture was used to assess the quality of data generated by each oversampling method. The MLP was trained with the Adam optimizer (learning rate: $1 \times 10^{-4}$, weight decay: $1 \times 10^{-3}$), and early stopping with a patience of 20 epochs. Validation AUC-ROC was monitored for binary tasks and micro-averaged AUC-ROC for multiclass tasks. Classifier details are in Appendix A.3.

**Evaluation Metrics:** For binary tasks, performance was assessed using AUC-ROC, Area Under the Precision-Recall Curve (AUPRC), and F1-score. For multiclass tasks, these metrics were reported with both macro and micro averaging (macro/micro AUC-ROC, macro/micro AUPRC, macro/micro F1-score) to capture overall and per-class performance.

## 4 RESULTS

### 4.1 PERFORMANCE ON TCGA LUNG CANCER DATASET (BINARY CLASSIFICATION)

The LEO-CVAE model trained successfully, exhibiting stable convergence. As detailed in Appendix A.4, the validation loss closely tracked the training loss across all components (total, KL divergence, and reconstruction), indicating no significant overfitting. Furthermore, the KL divergence remained well above the posterior collapse threshold, and the reconstruction correlation approached 1.0, confirming a well-regularized and effective generative model.

The comparative classification results are presented in Table 1. LEO-CVAE achieved the highest AUC-ROC ($0.661 \pm 0.030$) and AUPRC ($0.889 \pm 0.021$), indicating improved ranking ability and a better precision–recall trade-off for the minority class compared with all other methods. The No Oversampling baseline obtained the highest reported F1-score ($0.903 \pm 0.006$).

Non-generative oversampling strategies, such as Borderline-SMOTE and ADASYN, showed modest increases in AUPRC compared to the baseline, but these were often accompanied by minimal or no gains in AUC-ROC, reflecting a common trade-off in which minority-class precision improves at the expense of overall discriminative ability. In contrast, CVAE-based approaches generally outperformed non-generative methods, with LEO-CVAE standing out as the only method to deliver notable, simultaneous gains in both AUC-ROC and AUPRC over the baseline.

### 4.2 PERFORMANCE ON ADNI DATASET (MULTICLASS CLASSIFICATION)

This pattern of stable convergence was replicated on the multiclass ADNI dataset, confirming the model's reliability prior to performance evaluation (see Appendix A.4).

For the multiclass ADNI dataset (Table 2), the comparative results are more nuanced but again highlight the strengths of LEO-CVAE in achieving balanced performance across classes. The proposed model obtained the highest macro-averaged AUC-ROC ($0.587 \pm 0.025$), macro-AUPRC ($0.412 \pm 0.015$), and micro-AUPRC ($0.500 \pm 0.014$), indicating improved per-class discrimination and a stronger precision–recall trade-off. The No Oversampling baseline performed strongly on micro-averaged metrics, such as micro-AUC-ROC ($0.690 \pm 0.017$) and micro-F1 ($0.500 \pm 0.027$),

Table 1: Performance on the TCGA Lung Cancer Dataset

| Model | AUC-ROC | AUPRC | F1-Score |
|---|---|---|---|
| No Oversampling | $0.620 \pm 0.040$ | $0.868 \pm 0.028$ | $\mathbf{0.903} \pm 0.006$ |
| Random Oversampling | $0.613 \pm 0.065$ | $0.861 \pm 0.040$ | $0.767 \pm 0.021$ |
| SMOTE | $0.611 \pm 0.023$ | $0.868 \pm 0.025$ | $0.807 \pm 0.023$ |
| Borderline-SMOTE | $0.630 \pm 0.044$ | $0.874 \pm 0.022$ | $0.797 \pm 0.026$ |
| ADASYN | $0.617 \pm 0.052$ | $0.869 \pm 0.031$ | $0.786 \pm 0.026$ |
| Standard CVAE | $0.614 \pm 0.074$ | $0.869 \pm 0.040$ | $0.883 \pm 0.015$ |
| CVAE + Focal Loss | $0.645 \pm 0.043$ | $0.885 \pm 0.016$ | $0.871 \pm 0.028$ |
| **LEO-CVAE** | $\mathbf{0.661} \pm 0.030$ | $\mathbf{0.889} \pm 0.021$ | $0.881 \pm 0.012$ |

*Note:* Mean $\pm$ standard deviation over 5 folds. Best results are in bold.

Table 2: Performance on the ADNI Dataset

| Model | AUC-ROC | | AUPRC | | F1-Score | |
|---|---|---|---|---|---|---|
| | **Macro** | **Micro** | **Macro** | **Micro** | **Macro** | **Micro** |
| No Oversampling | $0.570 \pm 0.033$ | $\mathbf{0.690} \pm 0.017$ | $0.398 \pm 0.022$ | $0.492 \pm 0.021$ | $0.311 \pm 0.026$ | $\mathbf{0.500} \pm 0.027$ |
| Random Oversampling | $0.564 \pm 0.029$ | $0.588 \pm 0.020$ | $0.394 \pm 0.027$ | $0.393 \pm 0.018$ | $\mathbf{0.381} \pm 0.012$ | $0.402 \pm 0.020$ |
| SMOTE | $0.561 \pm 0.041$ | $0.606 \pm 0.030$ | $0.403 \pm 0.035$ | $0.416 \pm 0.035$ | $0.377 \pm 0.043$ | $0.410 \pm 0.040$ |
| Borderline-SMOTE | $0.558 \pm 0.049$ | $0.609 \pm 0.030$ | $0.392 \pm 0.040$ | $0.409 \pm 0.027$ | $0.373 \pm 0.037$ | $0.417 \pm 0.028$ |
| ADASYN | $0.556 \pm 0.037$ | $0.612 \pm 0.032$ | $0.383 \pm 0.034$ | $0.420 \pm 0.026$ | $0.367 \pm 0.042$ | $0.431 \pm 0.021$ |
| Standard CVAE | $0.563 \pm 0.032$ | $0.665 \pm 0.027$ | $0.386 \pm 0.025$ | $0.469 \pm 0.038$ | $0.359 \pm 0.024$ | $0.474 \pm 0.033$ |
| CVAE + Focal Loss | $0.562 \pm 0.027$ | $0.667 \pm 0.022$ | $0.395 \pm 0.029$ | $0.485 \pm 0.039$ | $0.347 \pm 0.029$ | $0.469 \pm 0.030$ |
| **LEO-CVAE** | $\mathbf{0.587} \pm 0.025$ | $0.683 \pm 0.013$ | $\mathbf{0.412} \pm 0.015$ | $\mathbf{0.500} \pm 0.014$ | $0.375 \pm 0.043$ | $0.484 \pm 0.020$ |

*Note:* Mean $\pm$ standard deviation over 5 folds. Best results are in bold.

which weight performance by class size and are often dominated by populous classes. In contrast, LEO-CVAE's superior macro-averaged metrics (unweighted averages that treat all classes equally) highlight its ability to distribute performance gains more evenly, benefiting minority classes without sacrificing overall discrimination.

CVAE-based methods generally delivered stronger micro-metrics, likely due to generating a more diverse set of synthetic samples, but LEO-CVAE distinguished itself with a balanced macro/micro profile.

## 5 ABLATION STUDY

We conducted a comprehensive ablation study to evaluate the contributions of LEO-CVAE's three core mechanisms by systematically disabling one or more of them: (i) the entropy-weighted loss, (ii) entropy-guided sampling, and (iii) inverse frequency class weighting. The specific configurations for each ablated model variant are detailed in Appendix A.5.

### 5.1 RESULTS AND ANALYSIS

The results for the binary TCGA lung cancer dataset are reported in Table 3, and for the multiclass ADNI dataset in Table 4. Across both datasets, the full LEO-CVAE consistently achieved balanced, high performance, demonstrating the value of integrating all three mechanisms.

On the TCGA lung cancer dataset (Table 3), the full LEO-CVAE achieved the highest AUC-ROC ($0.661 \pm 0.030$) and AUPRC ($0.889 \pm 0.021$). Removing entropy-weighted loss (Ablation 1) or entropy-guided sampling (Ablation 2) reduced both metrics, confirming the utility of each entropy-based component. While the Standard CVAE achieved a high F1-score, this was accompanied by a comparatively low AUC-ROC.

On the ADNI dataset (Table 4), the full LEO-CVAE achieved the highest macro-averaged AUC ($0.587 \pm 0.025$), indicating balanced performance across all classes. Ablation 2 (no entropy-guided sampling) achieved the highest micro-averaged AUC ($0.686 \pm 0.019$) and macro F1-score ($0.379 \pm 0.013$), though differences with the full model were small and within standard deviations. This highlights the LEWL as an especially impactful component of the LEO-CVAE.

Table 3: Ablation Study Results on the TCGA Lung Cancer Dataset

| Model Variant | AUC-ROC | AUPRC | F1-Score |
|---|---|---|---|
| **LEO-CVAE (Full Model)** | **0.661** $\pm$ 0.030 | **0.889** $\pm$ 0.021 | 0.881 $\pm$ 0.012 |
| LEO-CVAE (w/o Entropy-Weighted Loss) | 0.600 $\pm$ 0.054 | 0.863 $\pm$ 0.036 | 0.861 $\pm$ 0.025 |
| LEO-CVAE (w/o Entropy-Guided Sampling) | 0.631 $\pm$ 0.032 | 0.875 $\pm$ 0.025 | 0.861 $\pm$ 0.026 |
| LEO-CVAE (w/o Class Weights) | 0.616 $\pm$ 0.062 | 0.873 $\pm$ 0.023 | 0.865 $\pm$ 0.030 |
| CVAE + Class Weights | 0.617 $\pm$ 0.026 | 0.865 $\pm$ 0.025 | 0.868 $\pm$ 0.027 |
| Standard CVAE | 0.614 $\pm$ 0.074 | 0.869 $\pm$ 0.040 | **0.883** $\pm$ 0.015 |

*Note:* Mean $\pm$ standard deviation over 5 folds. Best results are in bold.

Table 4: Ablation Study Results on the ADNI Dataset

| Model Variant | AUC-ROC | | AUPRC | | F1-Score | |
|---|---|---|---|---|---|---|
| | Macro | Micro | Macro | Micro | Macro | Micro |
| **LEO-CVAE (Full Model)** | **0.587** $\pm$ 0.025 | 0.683 $\pm$ 0.013 | **0.412** $\pm$ 0.015 | **0.500** $\pm$ 0.014 | 0.375 $\pm$ 0.043 | 0.484 $\pm$ 0.020 |
| LEO-CVAE (w/o Entropy-Weighted Loss) | 0.544 $\pm$ 0.050 | 0.655 $\pm$ 0.031 | 0.380 $\pm$ 0.033 | 0.459 $\pm$ 0.043 | 0.322 $\pm$ 0.037 | 0.446 $\pm$ 0.031 |
| LEO-CVAE (w/o Entropy-Guided Sampling) | 0.579 $\pm$ 0.040 | **0.686** $\pm$ 0.019 | **0.412** $\pm$ 0.016 | 0.491 $\pm$ 0.023 | **0.379** $\pm$ 0.013 | **0.495** $\pm$ 0.025 |
| LEO-CVAE (w/o Class Weights) | 0.533 $\pm$ 0.019 | 0.646 $\pm$ 0.013 | 0.377 $\pm$ 0.021 | 0.445 $\pm$ 0.018 | 0.350 $\pm$ 0.034 | 0.458 $\pm$ 0.030 |
| CVAE + Class Weights | 0.566 $\pm$ 0.040 | 0.667 $\pm$ 0.031 | 0.387 $\pm$ 0.036 | 0.478 $\pm$ 0.040 | 0.322 $\pm$ 0.034 | 0.458 $\pm$ 0.044 |
| Standard CVAE | 0.563 $\pm$ 0.032 | 0.665 $\pm$ 0.027 | 0.386 $\pm$ 0.025 | 0.469 $\pm$ 0.038 | 0.359 $\pm$ 0.024 | 0.474 $\pm$ 0.033 |

*Note:* Mean $\pm$ standard deviation over 5 folds. Best results are in bold.

In summary, for the binary TCGA lung cancer dataset, all three components contribute to robust gains in AUC and AUPRC, with the entropy-based mechanisms having the largest impact. In the multiclass ADNI setting, the LEWL emerges as particularly effective, while removing entropy-guided sampling (Ablation 2) does not substantially harm performance and, in some cases, even slightly improves certain metrics. This validates the overall framework design by underscoring the power of the core entropy-weighted loss, while also highlighting that the optimal configuration of complementary components, such as class weighting and entropy-guided sampling, can be application-dependent.

## 6 CONCLUSION

In this work, we addressed the critical challenge of class imbalance in complex tabular data, with a focus on its application to clinical genomics data. Traditional oversampling methods, which rely on local, linear interpolation, are often ill-suited for such data. Their core assumption, that a straight line between two minority samples represents plausible data, frequently fails within the complex, non-linear manifolds characteristic of the biological processes underlying genomics data. While deep generative models like the Conditional Variational Autoencoder (CVAE) can capture these complex global distributions, a key limitation of standard CVAEs is that they learn the global distribution of a class by implicitly treating all training samples as equally informative. This uniform approach overlooks a critical insight: not all samples are equally valuable for refining a classifier's decision boundary. Instances located deep within a class's feature space are less informative than those situated in regions of high class overlap, where the boundary is most ambiguous. These "hard-to-learn" or "borderline" samples are strategically crucial, as they provide the most challenging examples for a classifier. While heuristic methods like Borderline-SMOTE and ADASYN pioneered the strategy of focusing on such instances, a significant gap remains in integrating a formal principle of uncertainty-awareness into a powerful distributional learning framework.

To bridge this gap, we introduce the Local Entropy-Guided Oversampling with a CVAE (LEO-CVAE), a novel generative oversampling framework that quantifies sample-level uncertainty using local Shannon entropy. By synergistically integrating this signal through a Local Entropy-Weighted Loss (LEWL) and an entropy-guided sampling strategy, LEO-CVAE focuses both its learning and generative processes on the most informative, class-overlapping regions of the feature space. Our empirical evaluation demonstrated that LEO-CVAE achieves superior and more balanced performance on challenging genomics datasets, delivering notable gains in AUC-ROC and AUPRC. The ablation study further revealed that the LEWL was the most impactful component, underscoring the benefit of compelling the model to learn a more robust representation of the contested decision boundary.

## 6.1 Limitations and Future Directions

This work lays the foundation for a new uncertainty-aware generative framework for imbalanced learning. Our study's focus on clinical genomics data was deliberate, as our framework is specifically designed for data with complex, non-linear characteristics. The principles of LEO-CVAE may also be well-suited for other high-dimensional omics data, such as proteomics and metabolomics, which share similar characteristics of complex, non-linear relationships. For simpler tabular data lacking these complex characteristics, traditional methods may still perform adequately. Future work should therefore extend this evaluation to a wider variety of tabular datasets, accompanied by formal statistical significance testing, to better characterize the regimes where different oversampling strategies are most effective.

Furthermore, several promising research avenues remain open. The core concepts of entropy-guided learning and generation could be generalized to other data modalities, such as images, or adapted for other generative architectures like GANs. One particularly promising direction could be adapting the LEO framework to Denoising Diffusion Models. Applying it to diffusion-based synthesis could amplify their strong generative capabilities: the denoising loss ($L_{DDPM}$) may be reweighted by the Local Entropy Score (LES) to counter majority-class gradient domination, while LES could also modulate class-conditional guidance during reverse diffusion, steering generation toward higher-quality minority samples. This integration of uncertainty-guided sampling with state-of-the-art synthesis offers a compelling path forward. Research also suggests that hybrid strategies combining traditional and generative models may yield further improvements (Wang et al., 2025a).

Finally, the mechanism for calculating local entropy warrants further investigation. Our choice of k-NN with Euclidean distance was a direct extension of the principles in SMOTE (Chawla et al., 2002), providing a robust baseline. However, the effectiveness of this uncertainty metric could be enhanced by exploring alternative distance metrics better suited for high-dimensional data, such as Manhattan distance and cosine similarity, or by moving beyond distance-based methods with techniques like Kernel Density Estimation (KDE). A particularly novel direction would be to leverage the probability distributions from a pre-trained model to guide the entropy calculation, potentially creating a more nuanced, model-aware measure of uncertainty.

## Ethics Statement

This research utilizes publicly available, de-identified patient data from The Cancer Genome Atlas (TCGA) and the Alzheimer's Disease Neuroimaging Initiative (ADNI) consortia. All data were originally collected with informed consent, and our study adhered to the data use policies of both repositories. The central goal of our work is to develop a method that mitigates classification bias against underrepresented minority classes in imbalanced datasets, a key challenge in promoting fairness in medical machine learning. The authors declare no conflicts of interest.

## Reproducibility Statement

To ensure the reproducibility of our results, we have made our data sources, experimental configurations, and code fully transparent.

Data: The TCGA lung cancer data can be accessed via the UCSC Xena platform (`https://xenabrowser.net/`). The ADNI dataset is available to qualified researchers upon application through the Laboratory of Neuro Imaging (LONI) website (`https://adni.loni.usc.edu/`). Access to both datasets is subject to the data use agreements of their respective repositories.

Model Architectures and Configurations: Complete details for our model architectures (Appendix A.2), the downstream classifier (Appendix A.3), and all baseline model configurations (Appendix A.1) are provided in the appendix.

Code: The source code for the LEO-CVAE framework is provided in the supplementary material.

## Acknowledgments

Gene expression (RNA-Seq) and corresponding clinical data for lung adenocarcinoma (LUAD) and lung squamous cell carcinoma (LUSC) were obtained from TCGA via the UCSC Xena.

Data collection and sharing for this project was also funded by the Alzheimer's Disease Neuroimaging Initiative (ADNI) (National Institutes of Health Grant U01 AG024904) and DOD ADNI (Department of Defense award number W81XWH-12-2-0012). ADNI is funded by the National Institute on Aging, the National Institute of Biomedical Imaging and Bioengineering, and through generous contributions from the following: AbbVie, Alzheimer's Association; Alzheimer's Drug Discovery Foundation; Araclon Biotech; BioClinica, Inc.; Biogen; Bristol-Myers Squibb Company; CereSpir, Inc.; Cogstate; Eisai Inc.; Elan Pharmaceuticals, Inc.; Eli Lilly and Company; EuroImmun; F. Hoffmann-La Roche Ltd and its affiliated company Genentech, Inc.; Fujirebio; GE Healthcare; IXICO Ltd.; Janssen Alzheimer Immunotherapy Research & Development, LLC.; Johnson & Johnson Pharmaceutical Research & Development LLC.; Lumosity; Lundbeck; Merck & Co., Inc.; Meso Scale Diagnostics, LLC.; NeuroRx Research; Neurotrack Technologies; Novartis Pharmaceuticals Corporation; Pfizer Inc.; Piramal Imaging; Servier; Takeda Pharmaceutical Company; and Transition Therapeutics. The Canadian Institutes of Health Research is providing funds to support ADNI clinical sites in Canada. Private sector contributions are facilitated by the Foundation for the National Institutes of Health (www.fnih.org). The grantee organization is the Northern California Institute for Research and Education, and the study is coordinated by the Alzheimer's Therapeutic Research Institute at the University of Southern California. ADNI data are disseminated by the Laboratory for Neuro Imaging at the University of Southern California.

The authors acknowledge the use of Large Language Models in preparing this manuscript; a statement detailing their role is provided in Appendix A.6

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

## A APPENDIX

This supplement provides detailed information on the network architectures and hyperparameter settings used in our experiments to ensure full reproducibility.

### A.1 BASELINE MODEL DESCRIPTIONS AND CONFIGURATIONS

We compared LEO-CVAE against seven baseline and alternative methods:

1. **No Oversampling**: The baseline performance of the classifier on the original, imbalanced data.

2. **Random Oversampling (Japkowicz, 2000)**: The simplest approach, which randomly duplicates samples from the minority class.

3. **SMOTE (Chawla et al., 2002)**: The classic Synthetic Minority Over-sampling Technique that generates new samples via linear interpolation.

4. **Borderline-SMOTE (Han et al., 2005)**: An advanced SMOTE variant that focuses synthetic sample generation on minority instances near the class boundary.

5. **ADASYN (Haibo et al., 2008)**: An adaptive approach that generates more synthetic data for minority samples that are harder to learn based on their local distribution.

6. **Standard CVAE (Sohn et al., 2015)**: A baseline CVAE model with an identical architecture to LEO-CVAE but trained using a standard, unweighted reconstruction loss.

7. **CVAE with Focal Loss (Lin et al., 2017)**: A CVAE trained with a focal loss-inspired reconstruction objective to focus learning on samples with high reconstruction error.

Non-generative baselines were implemented using the `imbalanced-learn` Python library. All CVAE-based baselines use the identical network architecture as LEO-CVAE (Supplement A.2). Specific configurations are in Table 5.

Table 5: Baseline Model Hyperparameters

| Model | Library/Base | Key Parameter | Value |
|---|---|---|---|
| **SMOTE** | `imbalanced-learn` | 'k_neighbors' | 7 (Binary) / 15 (Multiclass) |
| **Borderline-SMOTE** | `imbalanced-learn` | 'kind' | 'borderline-1' |
| | | 'k_neighbors' | 7 (Binary) / 15 (Multiclass) |
| | | 'm_neighbors' | 7 (Binary) / 15 (Multiclass) |
| **ADASYN** | `imbalanced-learn` | 'n_neighbors' | 7 (Binary) / 15 (Multiclass) |
| **Standard CVAE** | Our implementation | KLD Weight ($\beta$) | 1.0 |
| | | Minimum KLD | 0.1 |
| **CVAE with Focal Loss** | Our implementation | KLD Weight ($\beta$) | 1.0 |
| | | Minimum KLD | 0.1 |
| | | Focusing Param. ($\gamma$) | 1.0 |

## A.2 LEO-CVAE ARCHITECTURE AND HYPERPARAMETERS

Our proposed LEO-CVAE model and all CVAE-based baselines share the identical network architecture detailed below, with an input dimension of 64.

- **Encoder Architecture**:
  1. Input: Concatenation of feature vector ($D_{in} = 64$) and one-hot class label ($C$), size = $64 + C$.
  2. 'Linear'($64 + C \rightarrow 64$) $\rightarrow$ 'LeakyReLU'($0.2$) $\rightarrow$ 'Dropout'($p = 0.1$).
  3. 'Linear'($64 \rightarrow 32$) $\rightarrow$ 'LeakyReLU'($0.2$) $\rightarrow$ 'Dropout'($p = 0.1$).
  4. Two parallel 'Linear' output heads from the 32-neuron layer:
     - 'Linear'($32 \rightarrow 16$) for the latent mean ($\mu$).
     - 'Linear'($32 \rightarrow 16$) for the latent log-variance ($\log \sigma^2$).

- **Decoder Architecture**:
  1. Input: Concatenation of latent vector ($D_z = 16$) and one-hot class label ($C$), size = $16 + C$.
  2. 'Linear'($16 + C \rightarrow 32$) $\rightarrow$ 'LeakyReLU'($0.2$) $\rightarrow$ 'Dropout'($p = 0.1$).
  3. 'Linear'($32 \rightarrow 64$) $\rightarrow$ 'LeakyReLU'($0.2$) $\rightarrow$ 'Dropout'($p = 0.1$).
  4. 'Linear' output layer mapping from $64 \rightarrow 64$ to reconstruct the original feature vector.

The hyperparameters for training LEO-CVAE are detailed in Table 6.

Table 6: LEO-CVAE Training Hyperparameters

| Hyperparameter | Symbol | Value | Description |
|---|---|---|---|
| Optimizer | - | Adam | - |
| Learning Rate | - | $1 \times 10^{-3}$ | - |
| Weight Decay | - | $1 \times 10^{-5}$ | - |
| Batch Size | - | 32 | - |
| Max Epochs | - | 500 | - |
| Early Stopping Patience | - | 25 | - |
| Gradient Clip Norm | - | 1.0 | Maximum norm for gradient clipping. |
| Latent Dimension | $D_z$ | 16 | Dimensionality of the latent space. |
| k-nearest neighbors | $k$ | 7 (Binary) / 15 (Multiclass) | Neighbors for local entropy calculation. |
| Focus | $\gamma$ | 0.5 (Binary) / 2.5 (Multiclass) | Weighting for high-entropy samples. |
| KLD Weight | $\beta$ | 4.0 | Weight of the KL divergence term. |
| Minimum KLD | - | 0.1 | Floor to prevent posterior collapse. |

## A.3 MLP CLASSIFIER ARCHITECTURE

The Multi-Layer Perceptron (MLP) used as the downstream classifier has an input dimension of 64 and the following single-hidden-layer architecture:

- **Hidden Layer:** 'Linear' layer $(64 \rightarrow 32) \rightarrow$ 'BatchNorm1d' $\rightarrow$ 'ReLU' $\rightarrow$ 'Dropout' $(p = 0.5)$.
- **Output Layer:** 'Linear' layer mapping from $32 \rightarrow 1$ (Binary) or $32 \rightarrow 3$ (Multiclass).

The MLP was trained using the hyperparameters listed in Table 7.

Table 7: MLP Classifier Training Hyperparameters

| Hyperparameter | Value |
|---|---|
| Optimizer | Adam |
| Learning Rate | $1 \times 10^{-4}$ |
| Weight Decay | $1 \times 10^{-3}$ |
| Batch Size | 32 |
| Max Epochs | 200 |
| LR Scheduler | ReduceLROnPlateau |
| LR Scheduler Patience | 5 |
| LR Scheduler Factor | 0.7 |
| Early Stopping Patience | 20 |
| Early Stopping Metric | AUC (Binary) / AUC Micro (Multiclass) |
| Gradient Clip Norm | 0.5 |
| Label Smoothing | 0.1 |

## A.4   TRAINING HISTORY

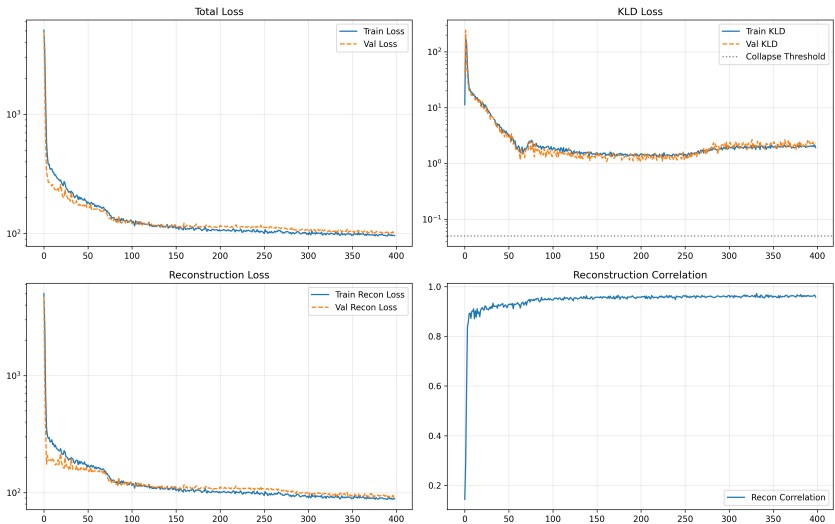

Figure 1: Training history of the LEO-CVAE model for a representative fold on the TCGA lung cancer dataset.

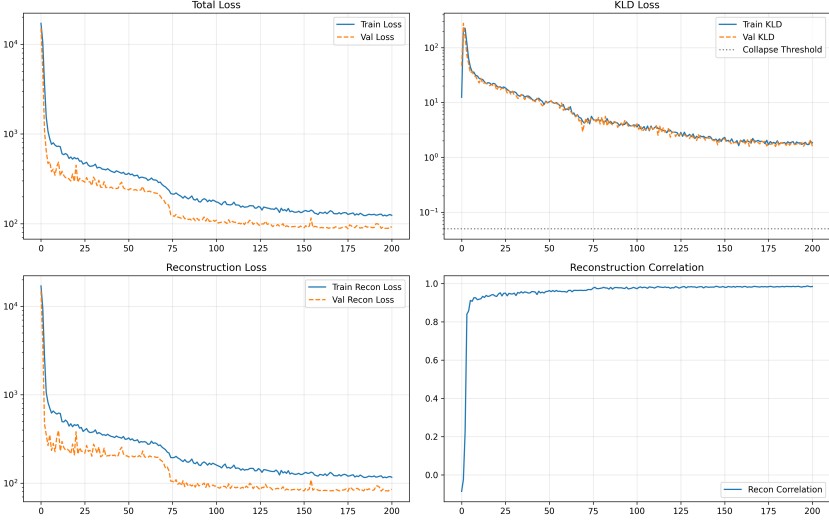

Figure 2: Training history of the LEO-CVAE model for a representative fold on the ADNI dataset.

## A.5 Ablation Study Design

Table 8: Ablation Study Experimental Design

| Model Variant | Entropy-Weighted Loss | Entropy-Guided Sampling | Class Weighting | Purpose |
|---|:---:|:---:|:---:|:---:|
| Full LEO-CVAE | ✓ | ✓ | ✓ | The complete proposed model. |
| LEO-CVAE (w/o Ent-Weighted Loss) | ✗ | ✓ | ✓ | Isolates the effect of the entropy-weighted loss. |
| LEO-CVAE (w/o Ent-Guided Sampling) | ✓ | ✗ | ✓ | Isolates the effect of entropy-guided sampling. |
| LEO-CVAE (w/o Class Weights) | ✓ | ✓ | ✗ | Isolates the effect of class weighting. |
| CVAE + Class Weights | ✗ | ✗ | ✓ | A simpler, non-entropy based model. |
| Standard CVAE | ✗ | ✗ | ✗ | The foundational generative baseline. |

## A.6 Statement on Large Language Model (LLM) Usage

The authors acknowledge the use of a large language model (Google's Gemini) as an assistant in the preparation of this manuscript. The LLM's role was in the writing and editing process, and included: (i) improving grammar, clarity, and conciseness throughout the paper; (ii) rephrasing and restructuring paragraphs and sections to enhance narrative flow and meet page-limit constraints; and (iii) assisting with LaTeX formatting and compliance checks against the conference style guidelines. The LLM was not used for research ideation, experimental design, data analysis, or drawing the scientific conclusions presented. All core intellectual contributions, including the proposal of the LEO-CVAE framework and the interpretation of results, are solely those of the human authors.

