# OpenReview forum: "Uncertainty-Aware Generative Oversampling Using an Entropy-Guided Conditional Variational Autoencoder"
_ICLR.cc/2026/Conference — ICLR 2026 Conference Withdrawn Submission_

### Official Review · Reviewer_rQPj · 2025-10-28

**Soundness:** 2
**Presentation:** 2
**Contribution:** 2
**Rating:** 4
**Confidence:** 3

**Summary:**

This paper proposes an uncertainty-aware generative oversampling framework, LEO-CVAE, to address class imbalance in high-dimensional and complex tabular data environments, focusing on clinical genomics datasets. The key innovation lies in using local Shannon entropy as a quantitative measure of sample-level uncertainty, guiding both training (via a Local Entropy-Weighted Loss, LEWL) and data generation (through entropy-guided sampling) in a conditional variational autoencoder (CVAE) setting.

**Strengths:**

1. Class imbalance is indeed a persistent challenge in medical AI, and the authors' choice of clinical genomics scenarios holds genuine practical value. The authors clearly articulate why existing methods fall short: SMOTE-based methods rely on linear assumptions that break down in high-dimensional nonlinear data, while standard CVAEs overlook the varying importance of different samples.
2. The method is clear and easy to follow. The authors introduce a single interpretable uncertainty measure (LES) and integrate it into two stages: training via entropy-weighted loss and generation via entropy-guided sampling, without modifying the CVAE backbone. The hyperparameters are few and intuitive.
3. The evaluation includes seven well-chosen baselines covering both traditional oversampling techniques (SMOTE variants, ADASYN) and generative alternatives (standard CVAE, CVAE with Focal Loss).

**Weaknesses:**

1. Only two datasets are used, both from the biomedical/genomics space and both pre-filtered to 64 features post-selection. While arguments for focusing on difficult, heterogeneity-rich data are credible, the lack of results on more generic/less complex tabular datasets (even as negative results) makes it difficult to assess practical generality.
2. While standard metrics are used, Tables 1 and 2 indicate that LEO-CVAE sometimes lags behind the No Oversampling baseline on micro-averaged metrics or F1. The paper notes these cases but offers limited analysis. A clearer discussion of why gains in macro metrics can coincide with drops in micro or overall F1 would strengthen the claims.
3. The paper notes the entropy focus hyperparameter $\gamma$ is set differently for binary (0.5) vs. multiclass (2.5) classification but gives no substantive rationale or sensitivity analysis.
4. Although the paper claims that LEO-CVAE produces more robust and plausible synthetic samples in overlap regions, the manuscript does not provide supporting visualizations. Plots of latent embeddings or generated samples using t-SNE or UMAP, together with real data overlays, would make the claim more convincing.
5. While the paper compares against seven baselines including traditional methods (SMOTE variants) and generative models (CVAE, CVAE+Focal Loss), the comparison is limited to relatively established techniques. More recent advances in generative modeling for imbalanced tabular data, such as diffusion-based approaches, metric learning-based generative methods, or ensemble generative strategies, are not discussed or compared.

**Questions:**

1. How sensitive is the framework to the choice of the entropy focus parameter $\gamma$?
2. Can the authors provide visualizations (e.g., t-SNE/UMAP) of real vs. synthetic samples before and after oversampling, specifically focusing on high-entropy or class-overlap regions?
3. Table 2 shows the No Oversampling baseline outperforms LEO-CVAE on micro-F1 (0.500 vs. 0.484) and micro-AUC-ROC (0.690 vs. 0.683). Can the authors clarify how they weigh macro vs. micro metric trade-offs and justify when degraded overall accuracy is acceptable for minority-class gains?

---

### Official Review · Reviewer_5F8F · 2025-10-30

**Soundness:** 3
**Presentation:** 1
**Contribution:** 2
**Rating:** 2
**Confidence:** 3

**Summary:**

This paper focuses on the problem of class imbalance in machine learning. Traditional oversampling approaches rely on linear interpolation and generate limited data diversity, while deep generative models treat all samples equally and overlook sample-level uncertainty. To address these limitations, this paper proposes an uncertainty-aware oversampling framework based on deep generative models. The proposed framework integrates local uncertainty, quantified using Shannon entropy, into the model training and data generation. Experiments conducted on two clinical genomics datasets demonstrate the enhanced model performance with the generated data, surpassing multiple generative baselines.

**Strengths:**

1. This paper addresses the problem of data imbalance in machine learning, which is a critical challenge that commonly exists in real-world applications.
2. The paper introduces an entropy-based score to measure sample-level uncertainty and incorporates it in both learning and generation processes in the proposed framework, offering an insightful and well-motivated perspective.

**Weaknesses:**

1. The paper would benefit from an overview figure illustrating the proposed learning and generation framework, which could make the methodology clearer and the data generation process more intuitive.
2. While the paper presents a solid method for data generation, the experimental setup is quite limited. The evaluation focuses only on tabular data, leaving it unclear how well the proposed method generalizes to other data types, such as medical imaging data.
3. The experimental baselines appear insufficient and lack recent methods for comparison. Additionally, the paper mentions several CVAE-based works in the Introduction section, such as DVAE, CTVAE, and MGVAE; however, these methods are not included in the empirical comparisons.
4. The paper emphasizes that the core novelty of the proposed method lies in integrating the uncertainty score into data generation. However, the entire experiments only report prediction performance metrics. No uncertainty-related metrics are provided to demonstrate how the proposed approach affects the uncertainty of the generated data.
5. The paper notes that previous methods are computationally intensive; however, no comparison of computational cost is reported.

**Questions:**

1. Could the paper discuss how the proposed method generalizes to image data and how it compares with modern diffusion models for data generation?
2. How would the paper evaluate the uncertainty of the generated data? Would common calibration metrics, such as Expected Calibration Error (ECE) and Adaptive Calibration Error (ACE) metrics, be suitable for assessing whether the proposed method generates more diverse data compared to baselines?
3. How sensitive is the proposed method to the choice of hyperparameters, such as $k$ in the $k$-nearest neighbors used for local entropy score calculation?

---

### Official Review · Reviewer_kLTi · 2025-11-02

**Soundness:** 2
**Presentation:** 3
**Contribution:** 2
**Rating:** 4
**Confidence:** 4

**Summary:**

This paper presents a generative model-based oversampling method to overcome class imbalance for downstream classification tasks. The method extends conditional VAE by introducing a local-entropy weighting term in the standard VAE loss, in addition to further upweighting as a function of imbalance. At sampling time, a seed data point is sampled weighted by its local entropy. The intention for both is to improve sampling of points around uncertain regions. The authors demonstrate that the proposed oversampling method improves MLP-based classifier on downstream binary and multiclass classification tasks using gene expression data and clinical labels.

Note that the manuscript is 9.5 pages including abstract, which is grounds for desk reject but I will provide my comments nevertheless.

**Strengths:**

- the paper clearly and effectively outlines the motivation, and is in general well-written and easy to follow
- the proposed method is simple and elegant and appears to work well, and I’m genuinely surprised it has not been proposed before as it is a very intuitively appealing idea
- the empirical experiments have sufficient baselines, and the ablation studies demonstrate the value of the proposed additions

**Weaknesses:**

- it remains unclear throughout in which space the nearest neighbors distances are computed, only mentioning, e.g., “to idetnfiy high-entropy regions within the feature space (line 127)”, and this is a critical detail
- the empirical evaluation is rather limited: even though it is tested on real data with multiple baselines, the performance of the proposed method is nevertheless likely to be a function of both the downstream classifier and the data. Does it improve the performance of simpler or more complex (including SOTA) classifiers on such transcriptomics data problem?
- similarly, a few simpler / more intuitive problems, such as artificially imbalanced mnist, would be more illustrative, as transcriptomics data have fairly specific data characteristics (discrete counts, many zeros, etc.)
- the paper is over page limit

**Questions:**

- line 166: the kNNs are found using Euclidean distances in feature space, what exactly are the features here? Does it mean latent space, i.e., distance in z? It would be informative already to mention this here.
- the kNNs in the proposed LES does not account for how far samples are in an absolute sense, so that the nearest neighbor of a sample in a very sparse region of the feature space could be very far away and of a different class, which would be fine from a local uncertainty point of view. Is this an issue?
- is oversampling performance a function of the classifier? The authors should demonstrate that the method consistently improves more than a single type of classifier

---

### Official Review · Reviewer_Rjgh · 2025-11-03

**Soundness:** 2
**Presentation:** 4
**Contribution:** 2
**Rating:** 4
**Confidence:** 3

**Summary:**

The paper tackles class imbalance in high-dimensional biomedical/tabular data by proposing LEO-CVAE, a generative oversampling framework that is uncertainty-aware. It computes a Local Entropy Score (LES)—the Shannon entropy of class labels in a sample’s k-nearest-neighbor neighborhood—to identify borderline/high-uncertainty regions. LES is used twice: (1) to reweight the CVAE reconstruction loss so the generator learns boundary regions better, and (2) to guide sampling so new synthetic samples are concentrated near those regions. The synthetic data are added to the training set, then a downstream classifier is trained. On the ADNI and TCGA datasets, LEO-CVAE demonstrates consistent gains over classic oversampling (SMOTE/ADASYN) and generative baselines.

**Strengths:**

1. **Relevant and clinically meaningful problem framing.**
The paper addresses class imbalance in high-dimensional medical and omics data, a setting where nonlinear structures and small-sample effects are particularly severe. The motivation is clear and carries practical clinical significance.

2. **Empirical validation on two real datasets.**
The method is tested on TCGA and ADNI datasets, covering both binary and multi-class clinical prediction tasks, which supports the potential generalizability of the approach.

3. **Comprehensive ablation study.**
The paper includes component-wise ablations (e.g., removing entropy weighting or entropy-guided sampling), which help clarify the contribution of each proposed module.

4. **Clear and structured method presentation.**
The proposed framework and training procedure are described clearly and systematically, making the paper relatively easy to follow and reproducible.

**Weaknesses:**

1. **Marginal gains without statistical support.**
Across both datasets, improvements in AUC and related metrics appear marginal, often smaller than the across-fold standard deviation. It is suggested to report paired statistical tests vs. the strongest baseline.

2. **Missing dataset-level analysis of local uncertainty (LES).**
Since the method relies on LES, it would be helpful to report the distributions of the local entropy scores and determine whether the method effectively helps classify cases with high LES.

3. **Lack of controlled synthetic studies to validate the mechanism.**
The paper would have been better if it had been evaluated in synthetic settings with tunable overlap and imbalance.

4.  **Clarify the premise of oversampling high-uncertainty regions: aleatoric vs. epistemic.**
If high LES primarily reflects aleatoric uncertainty (label ambiguity/noise), oversampling may not help and could amplify noise. There should be some experiments that relate LES to predictive uncertainty and label noise, and discussions on when entropy-guided oversampling is appropriate vs. when alternative strategies are preferable.

**Questions:**

See weaknesses.

---

### Note · Authors · 2026-01-25

**Comment:**

We are withdrawing this submission to incorporate further experimental results and refine the methodology. We thank the reviewers for their time and feedback.

**Withdrawal Confirmation:**

I have read and agree with the venue's withdrawal policy on behalf of myself and my co-authors.